# The causal relationship between gut microbiota and immune skin diseases: A bidirectional Mendelian randomization

**Fei Feng** [1], **Ruicheng Li** [2], **Rui Tian** [2], **Xueyi Wu** [2], **Nannan Zhang** [2], **Zhenhua Nie** [3] *

**1** Tianjin Medical University, Tianjin, China, **2** Tianjin University of Traditional Chinese Medicine, Tianjin, China, **3** Tianjin Academy of Traditional Chinese Medicine Affiliated Hospital, Tianjin, China

* niezhenhua@163.com

## Abstract

### Background

Increasing evidence suggests that alterations in gut microbiota are associated with a variety of skin diseases. However, whether this association reflects a causal relationship remains unknown. We aimed to reveal the causal relationship between gut microbiota and skin diseases, including psoriasis, atopic dermatitis, acne, and lichen planus.

### Methods

We obtained full genetic association summary data for gut microbiota, psoriasis, atopic dermatitis, acne, and lichen planus from public databases and used three methods, mainly inverse variance weighting, to analyze the causal relationships between gut microbiota and these skin diseases using bidirectional Mendelian randomization, as well as sensitivity and stability analysis of the results using multiple methods.

### Results

The results showed that there were five associated genera in the psoriasis group, seven associated genera were obtained in the atopic dermatitis group, a total of ten associated genera in the acne group, and four associated genera in the lichen planus group. The results corrected for false discovery rate showed that Eubacteriumfissicatenagroup (P = 2.20E-04, OR = 1.24, 95%CI:1.11–1.40) and psoriasis still showed a causal relationship. In contrast, in the reverse Mendelian randomization results, there was no evidence of an association between these skin diseases and gut microbiota.

### Conclusion

We demonstrated a causal relationship between gut microbiota and immune skin diseases and provide a new therapeutic perspective for the study of immune diseases: targeted modulation of dysregulation of specific bacterial taxa to prevent and treat psoriasis, atopic dermatitis, acne, and lichen planus.

**Data Availability Statement:** All relevant data are within the paper and its Supporting Information files.

**Funding:** The author(s) received no specific funding for this work.

**Competing interests:** The authors have declared that no competing interests exist.

## 1. Introduction

Skin is the largest immune organ of the human body, which protects the human body from external aggression. More and more studies have found that many skin diseases are related to the overall balance of the body, such as immune status, body metabolism level, gut microbiota homeostasis, etc. Psoriasis (PSO), atopic dermatitis (AD), acne, and lichen planus (LP) are common skin disorders in clinical practice. Psoriasis is an immune-related chronic inflammatory skin disease. Well-defined erythematous patches with silvery scales were seen on the skin. Numerous studies have shown that psoriasis is associated with autoimmune, metabolic, gastrointestinal, and mental health disorders [1–4]. Pruritus, pleomorphic lesions, and an exudate tendency are common characteristics of atopic dermatitis, a chronic relapsing and remitting inflammatory skin disease [5]. Atopic dermatitis has been linked to genetics, autoimmunity, environment, gastrointestinal health, and mental health. The etiology and pathogenesis of this disease are not very clear, and it is generally believed that it may be the result of the interaction between genetic factors and environmental factors through immune-mediated pathways [6]. In adolescents and young adults, acne is a common chronic inflammatory disease of the sebaceous glands [7]. In the past, the pathogenesis of acne has not been fully understood. Genetic factors, androgen-induced sebaceous secretion, follicular sebaceous gland duct keratosis, Propionibacterium carvings reproduction, immune inflammatory response, and other factors may be related to it. The pathogenesis of some patients is also affected by genetic, immune, endocrine, emotional, and dietary factors. Numerous studies have shown that gut microbiota imbalance plays a crucial role in the pathogenesis of acne [8]. The lichen planus is an inflammatory skin disease that affects the skin, mucosa, and adnexa [9]. It occurs on the flexed limbs. Typically, the lesions are elevated, purplish-red, flat papules, millet to green bean size or larger, polygonal or rounded, well circumvallated, with a ceroid film on the surface, and white shiny dots or fine light white reticular stripes. The etiology is still unclear and may be related to immune, genetic, viral infection, neuropsychiatric factors, and so on. The incidence of these common skin diseases has gradually increased in the past 30 years, which has become a global public health problem.

Gut microbiota (GM) is a large and complex microbial community in the intestine, which is considered to be closely related to the body's protection, immunity, metabolism, and nutrition. Gut microbiota not only directly affects the gut, but also potentially affects the normal physiology and homeostasis of other organs, such as the lung, brain, liver, and skin. In order to maintain gut-skin homeostasis, intestinal microbiota plays an important role [10]. When the relationship between gut microbiota and the immune system changes, it will have a certain impact on the skin, which may promote the occurrence and development of certain skin diseases.

A variety of skin disorders are associated with altered gut microbiota, according to some observational studies [11]. The pathogenesis of psoriasis, atopic dermatitis, acne, and lichen planus may be related to gut microbiota. However, the specific mechanism of the gut microbiota affecting skin health is not clear. To deepen our understanding of skin diseases, we applied Mendelian randomization to further explore the causal relationship between gut microbiota and psoriasis, atopic dermatitis, acne, and lichen planus.

Mendelian randomization(MR) is a method for exploring the causal relationship between exposure and outcomes based on pooled data from genome-wide association studies (GWAS) [12]. Causal relationships were inferred by selecting genetic variants significantly associated with exposure as instrumental variables [13]. This method has been widely used in the epidemiological study of some diseases. Compared with the traditional observational study, it has a good effect of removing confounding factors and makes the results more stable and reliable

[14, 15]. In this study, we used bidirectional two-sample Mendelian randomization (TSMR) analysis to investigate the possible causal relationship between gut microbiota and psoriasis, atopic dermatitis, lichen planus, and acne, which may provide a new idea for the study of these skin diseases: the treatment, control, and prevention of skin diseases by regulating gut microbiota.

## 2. Materials and methods

### 2.1 Data sources

GWAS data for gut microbiota were derived from the largest genome-wide meta-analysis of gut microbiota composition published to date by the MiBioGen consortium [16]. A total of 18,340 individuals were included in the study, most of whom had European ancestry (n = 13,266). GWAS was used to analyze the variation of GM taxa in different populations, and 122,110 variation points were obtained from 211 taxa (five levels from genus to phylum). We obtained data from the MiBioGen consortium(https://mibiogen.gcc.rug.nl/.) on a total of 131 genera at the genus level for this large-scale GWAS data. At the same time, 12 undefined genera were excluded, and 119 genera were finally used for this study. SNPs were quality checked to obtain the required IVs to ensure robustness and accuracy of the results: (1) Genome-wide significant SNPs associated with GM taxa were identified (P<5×10–8). Since the number of eligible IVS (P<5×$10^{-8}$) was minimal, a relatively more comprehensive threshold (P<1×$10^{-5}$) [17, 18] was selected in the literature for more comprehensive results. (2) A linkage disequilibrium (LD) analysis (R2<0.001, clumping distance = 10,000kb) was performed to satisfy the MR hypothesis. It is important to avoid weak instrument bias when inferring causal relationships, we used the formula F = $\beta^2$ exposure/$SE^2$ exposure to calculate the intensity of IVs [19–21] and excluded IVs with F < 10 (23). GWAS summary statistics for skin diseases were extracted from the FinnGen research project (https://r10.finngen.fi/), a gene-wide association study of various disease endpoints conducted on the same cohort in 2022, Its total sample size was 412,181 (230,310 females and 181,871 males), with an analyzed total of 21,311,942 variants and 2,408 available disease endpoints (phenotypes). psoriasis data included 6408 cases and 397564 controls, atopic dermatitis included 15208 cases and 367046 controls, acne included 3245 cases 394105, and lichen planus included 6411 cases and 405770 controls.

### 2.2 Statistical analysis

A primary method for calculating causal effect values, unbiased estimates were obtained by using the inverse variance weighting (IVW) method in the absence of horizontal polymorphism [22]. Additionally, the WM method and MR-Egger test were used in the MR analysis [23]. Heterogeneity-based analysis, IVW was tested using the fixed/random effects model. The OR and 95% confidence intervals (CIs) show the magnitude of the effect. Over 50% of SNPs are heterogeneous, and the result of WM is considered to be a significant causal effect value. If more than 50% of SNPs are polymorphic, the results for MR-Egger are still valid. The stability of results was analyzed by applying Cochrane's Q heterogeneity test, and IVs with P<0.05 were considered heterogeneous [24, 25]. For the purpose of ensuring the accuracy of the results for GM taxa causally associated with skin diseases (under the IVW), the MR-Egger intercept test and MR pleiotropy residual sum assessed the presence of potential polymorphisms in IVs using the (MR-PRESSO) global test (R package "MRPRESSO") Further horizontal pleiotropy tests were performed and possible outliers were removed. p<0.05 was considered as the presence of horizontal pleiotropy. Data robustness was also verified using the leave-one-out method [14].

To better evaluate the causal relationship between gut microbiota and diseases, reverse MR analysis was also performed with a disease as exposure and gut microbiota as an outcome. Based on the forward MR, the method and settings used were consistent.

R software (version 4.2.1) was used for all statistical analyses. A causal relationship between GM taxa and diseases was analyzed using the R package "TwoSampleMR". Statistical significance of P<0.05 was considered evidence of a potential causal effect [26–28]. The false discovery rate (FDR) was used to correct the results, and the FDR value < 0.1 was considered a robust result. It was considered suggestive when the P < 0.05 but the FDR ≥ 0.1.

### 2.3 Ethical approval

We used public de-identified GWAS data in this study. The ethics committee approved the data; therefore, additional ethical approval was not required.

## 3. Results

Based on the set criteria for IVs, we selected 1532 SNPs as IVs for 119 genera (F>10 for individual SNP), please refer to S1 Table for information on specific instrumental variables. Fig 1 shows the results of the MR analysis between the gut microbiota and the four skin diseases. Preliminary results showed that five genera were associated with at least one method in the psoriasis group, seven genera in the atopic dermatitis group, ten genera in the acne group, and four genera in the lichen planus group (Fig 1).

### 3.1 Pso

In our results, it was found that among the main IVW methods: Eubacteriumfissicatenagroup (P = 2.20E-04, OR = 1.24, 95%CI:1.11–1.40) were positively associated with psoriasis, while Dialister (P = 2.20E-02, OR = 0.79, 95%CI = 0.65–0.97), Terrisporobacter (P = 3.60E-02, OR = 0.81, 95%CI = 0.66–0.99) were inversely associated with psoriasis. In addition, LachnospiraceaeUCG008 (P = 4.70E-02, 0R = 2.23,95%CI: 1.14–4.38) was positively associated with psoriasis and RuminococcaceaeUCG002 (P = 6.10E-03, 0R = 0.57,95%CI: 0.40–0.82) was inversely associated with psoriasis in MR Egger's method (Fig 2). After FDR correction, Eubacteriumfissicatenagroup was still relevant (Table 1 and Fig 3). Cochrane's Q-test analysis of heterogeneity supports these results. Also MR-Egger, weighted median and MRPRESSO tests for selected IVs did not have horizontal pleiotropy (S2 Table). The MRPRESSO test results for Lachnospira (P = 3.50E-02, OR = 0.62, 95%CI: 0.40–0.97) revealed the presence of horizontal pleiotropy in the instrumental variables (IVs). After removing the outlier IVs, the MR results became non-significant (P>0.05).

### 3.2 AD

The IVW results showed RuminococcaceaeNK4A214group (P = 1.30E-02, 0R = 1.16,95%CI: 1.03–1.31), RuminococcaceaeUCG011 (P = 2.20E-02, OR = 1.09,95%CI = 1.01–1.17) were positively correlated with atopic dermatitis. Eubacteriumbrachygroup (P = 3.60E-02, 0R = 0.93,95%CI: 0.86–0.99), Dialister (P = 6.80E-03, 0R = 0.85,95%CI: 0.76–0.96), ChristensenellaceaeR.7group (P = 7.70E-03, 0R = 0.93,95%CI: 0.86–0.99), Intestinimonas (P = 2.30E-02, 0R = 0.90,95%CI: 0.82–0.98) were inverse correlations (Fig 2). The MR Egger method indicates a positive association between Ruminococcusgnavusgroup (P = 1.80E-02, 0R = 1.81,95%CI: 1.21–2.71) with AD. Cochrane's Q-test analysis supports these results. Furthermore, MR-Egger, weighted median, and MRPRESSO tests indicate no horizontal pleiotropy in the instrumental variables (IVs). The MRPRESSO test results for Faecalibacterium (P = 3.00E-02,

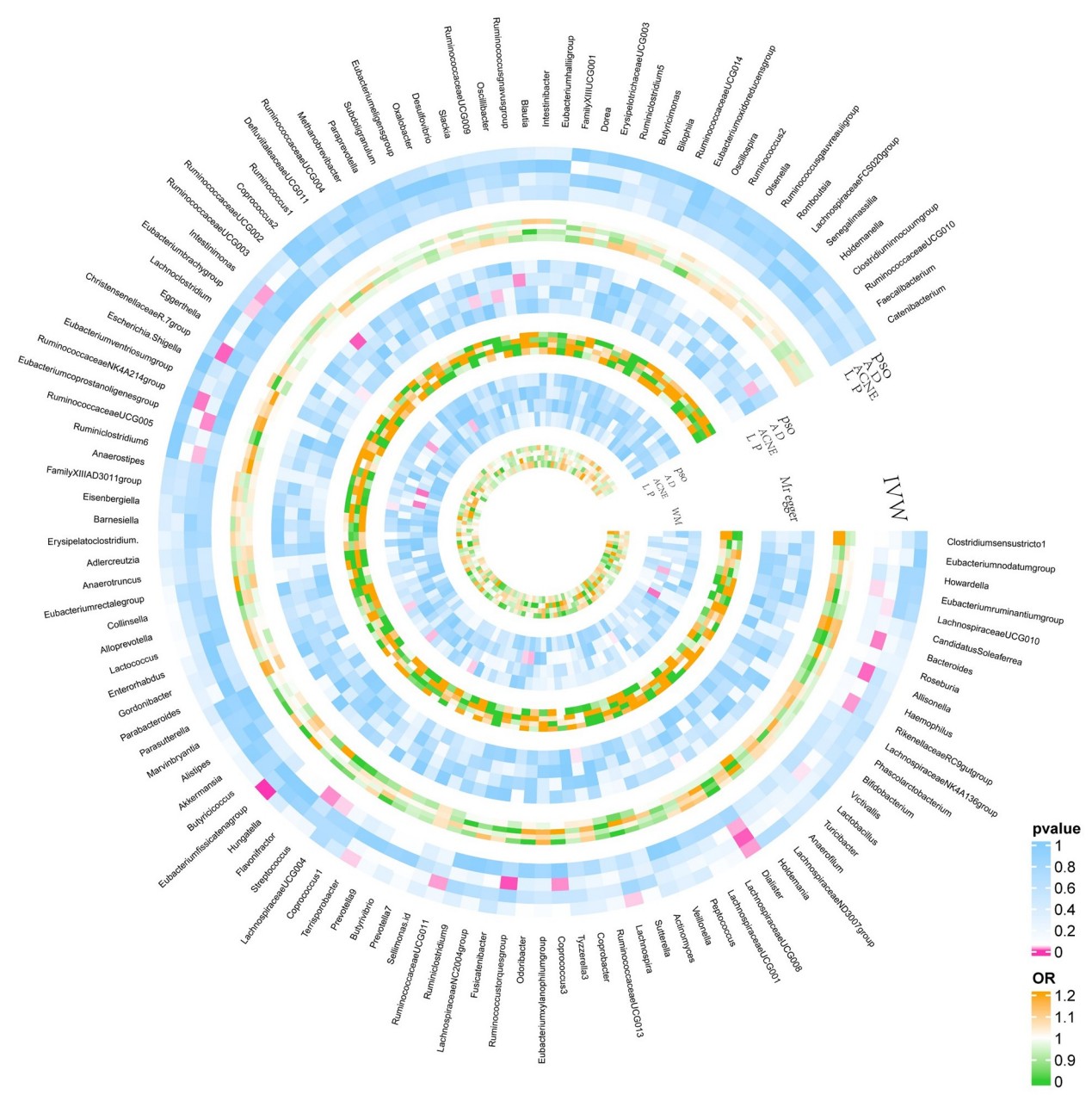

**Fig 1. Results of MR analysis between gut microbiota and PSO, AD, ACNE, LP.**

0R = 1.37,95%CI: 1.08–1.72) revealed the presence of horizontal pleiotropy in the instrumental variables (IVs). After removing the outlier IVs, the MR results became non-significant (P>0.05). No positive results were found after FDR correction (S3 Table).

### 3.3 Acne

Bacteroides (P = 1.40E-02, 0R = 1.57,95%CI: 1.09–2.24), Allisonella (p = 9.20E-03,OR = 1.24, 95%CI = 1.05–1.46), Coprococcus3(P = 1.80E-02,OR = 1.48, 95%CI = 1.07–2.06),

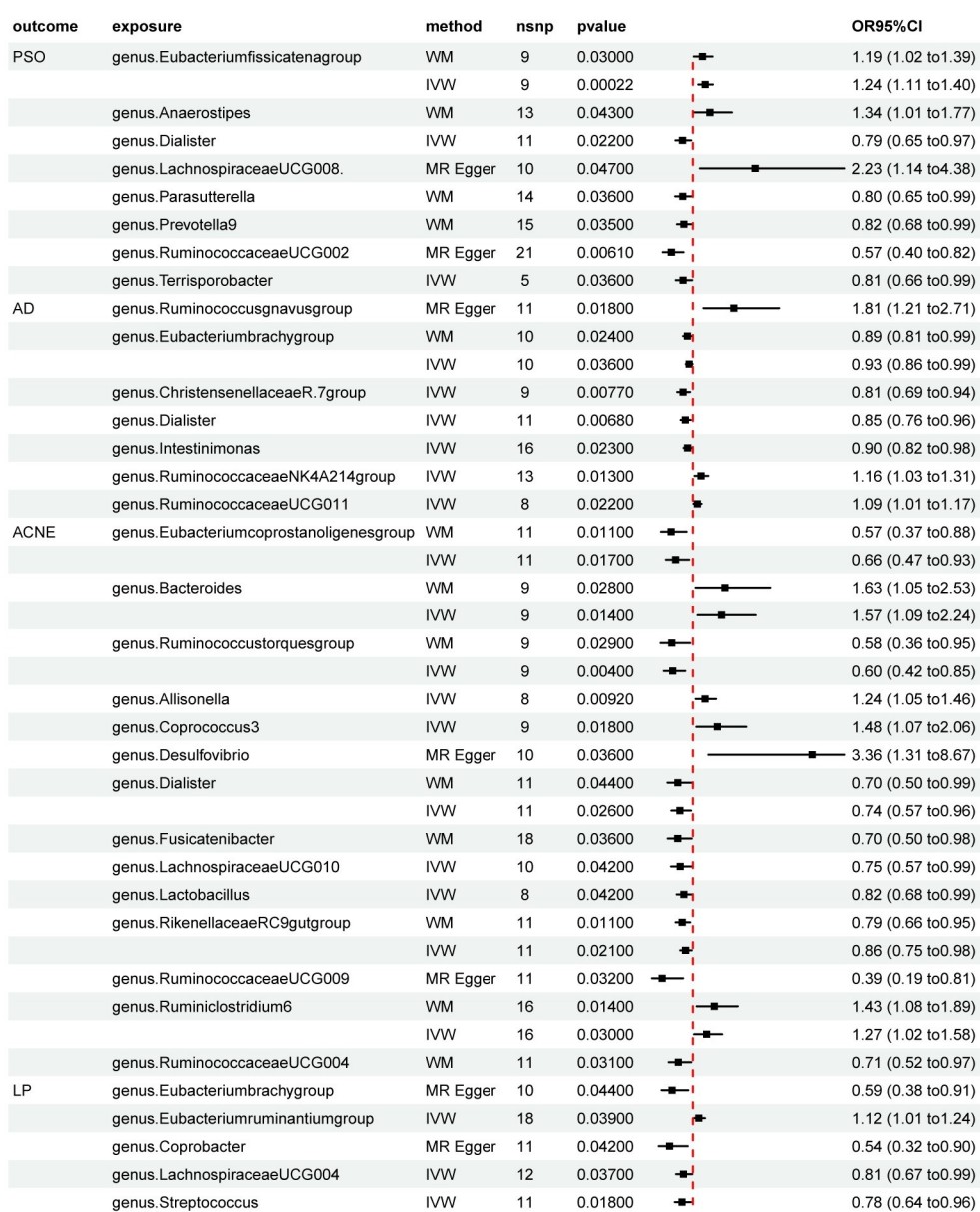

**Fig 2. Forest plots of MR analysis result between 26 genera and PSO, AD, ACNE, LP.**

Ruminiclostridium6 (P = 3.00E-02,OR = 1.27, 95%CI = 1.05–1.58) were positively correlated with acne, Fusicatenibacter (P = 1.60E-02, OR = 0.72, 95%CI = 0.55–0.94), Lactobacillus (P = 5.00E-03, OR = 0.75, 95%CI = 0.61–0.92), RikenellaceaeRC9gutgroup (P = 3.80E-02, OR = 0.86, 95%CI = 0.74–0.99) and Eubacteriumcoprostanoligenesgroup (P = 1.70E-02,

**Table 1. Table of Mendelian randomization results for a causal relationship between gut microbiota and skin diseases (P<0.05 & FDR<0.1).**

| Disease | Human gut microbiota | Method | nSNP | OR | 95%CI | P-value | FDR | Q(P-value) | Pleiotropy- P-value |
|---------|----------------------|--------|------|------|----------|----------|------|-------------|---------------------|
| | | IVW | 9 | 1.24 | 1.11–1.40 | 2.20E-04 | 0.026 | 6.50(0.59) | 0.84 |
| PSO | Eubacteriumfissicatenagroup | MR Egger | | 1.33 | 0.73–2.42 | 0.39 | 0.99 | 6.45(0.49) | |
| | | WM | | 1.19 | 1.02–1.39 | 0.03 | 0.92 | | |

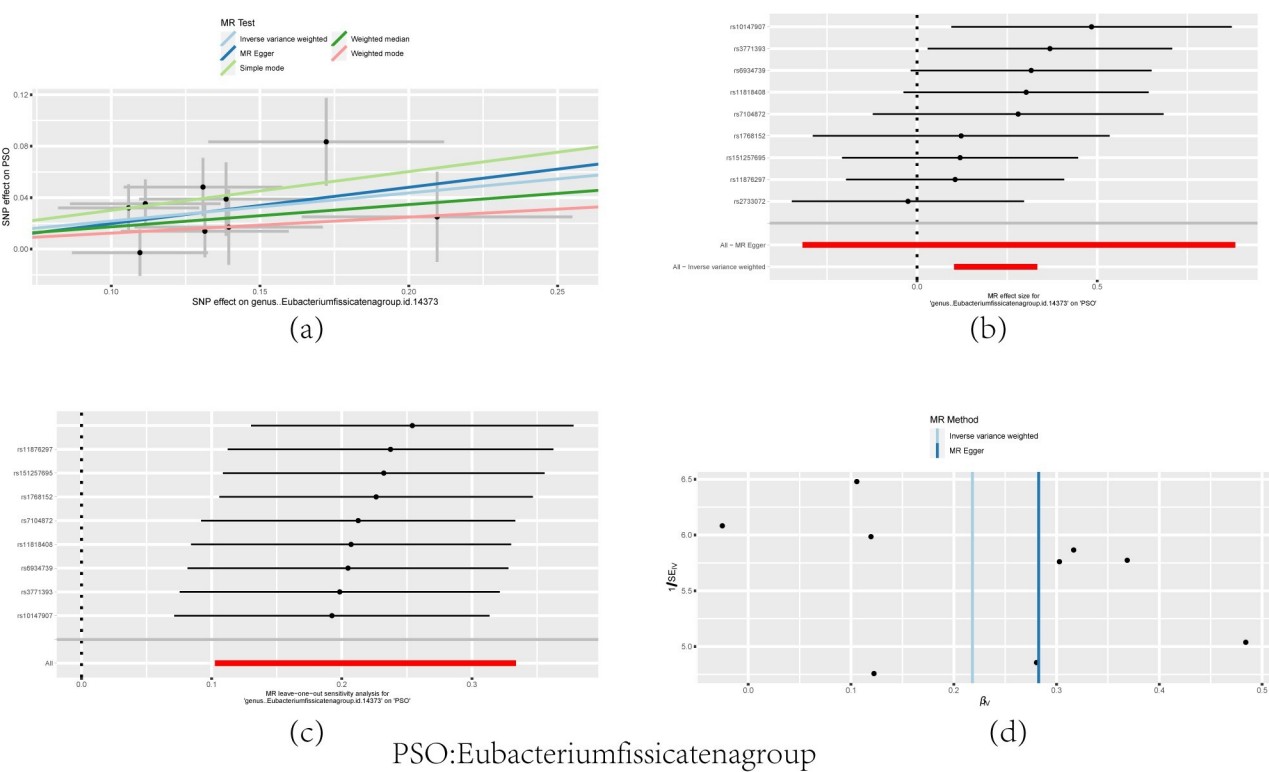

PSO:Eubacteriumfissicatenagroup

**Fig 3. Scatterplots and forest plots of MR analysis results between Eubacteriumfissicatenagroup and PSO, and examination of SNPs using leave-one-out and funnel plots.**

OR = 0.66, 95%CI = 0.47–0.93) shows a negative association. The MR Egger method indicates a positive association between Desulfovibrio (P = 3.60E-02, OR = 3.36, 95%CI = 1.31–8.67), with acne, while RuminococcaceaeUCG009 (P = 3.20E-02, OR = 0.39, 95%CI = 0.19–0.81) shows a negative association (Fig 2). After FDR correction, no positive results were found. Heterogeneity analysis of Cochrane's Q-test all showed good stability. Also MR-Egger, weighted median and MRPRESSO tests for selected IVs did not have horizontal pleiotropy (S4 Table).

### 3.4 LP

wEubacteriumruminantiumgroup (P = 3.90E-02,OR = 1.12, 95%CI = 1.01–1.24) was positively correlated with LP., LachnospiraceaeUCG004 (P = 3.70E-02,OR = 0.81, 95%CI = 0.67–0.99) and Streptococcus (P = 1.80E-02,OR = 0.78, 95%CI = 0.64–0.96) show a negative association, MR Egger showed that Coprobacter (P = 4.20E-02,OR = 0.54, 95%CI = 0.32–0.90) was negatively correlated(Fig 2). No positive result was found after FDR correction. The Cochrane's Q test for heterogeneity analysis supported these results. Also MR-Egger, weighted median and MRPRESSO tests for selected IVs did not have horizontal pleiotropy (S5 Table).

### 3.5 Reverse MR analysis

Considering the cross-validation with the forward results, the reverse MR analysis did not give the same results as the forward. Also, the FDR correction did not show any positive results.

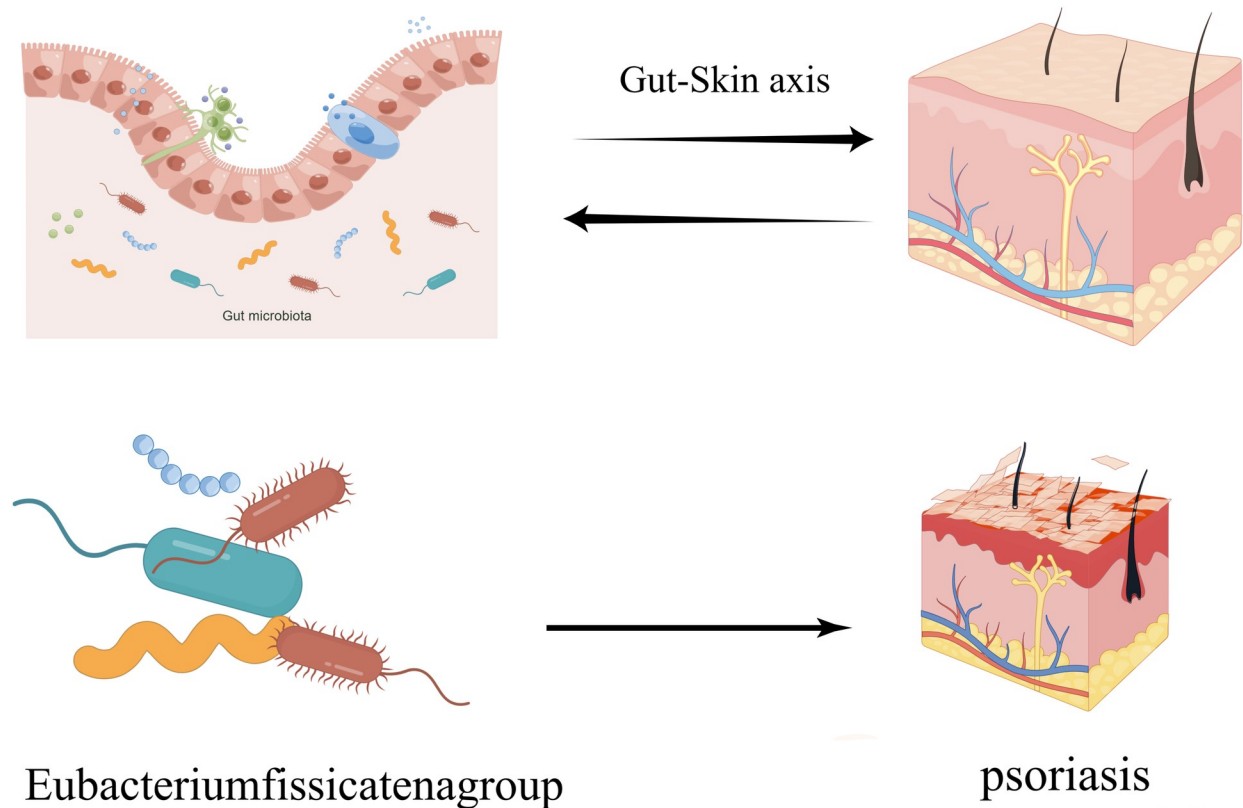

**Fig 4. Interaction between gut microbiota and skin status.** The relationship between gut microbiota and skin disease was obtained by MR analysis results (p<0.05&FDR<0.1).

## 4. Discussion

Using large-scale GWAS statistics, our current TSMR analysis identified 26 bacterial genera with possible causal associations with psoriasis, atopic dermatitis, lichen planus, and acne. Among them, there is strong evidence that Eubacteriumfissicatena are positively associated with psoriasis (Fig 4).

Skin-specific and immune-mediated, psoriasis is a chronic skin condition, mainly mediated by regulatory dendritic cells (DCs), Th17, and keratinocytes [29–31]. Associated with psoriasis, inflammatory bowel disease (IBD) increases inflammation through the development of Th17 cells in the gut microbiome, metabolites in the gut are altered as a result of changes in the gut microbiome. The increase in the content of oleic acid and stearic acid aggravates the psoriasis appearance by affecting Th17 cells [32, 33]. Patients with psoriasis and IBD have also been shown to have decreased abundances of beneficial microorganisms such as Akkermansia and Ruminoccocus. By competing with pathogens, these beneficial microbes prevent colonization by them. Therefore, their reduction leads to impaired intestinal barrier function, facilitating bacteria transfer from the gut to the systemic circulation in patients with psoriasis, resulting in altered immune responses and increased proinflammatory responses. Our study obtained similar results RuminococcaceaeUCG011 is a protective factor for psoriasis [34–36]. Moreover, an increase in the Lachnospiraceae UCG008 bacterial group, as well as a decrease in the Dialister and Terrisporobacter bacterial groups, may trigger or exacerbate psoriasis. After correction for FDR, our TSMR study also found that Eubacteriumfissicatena were still relevant.

The immune imbalance biased by Th2 and Th17 is an important pathogenesis of AD [37], the Treg cells are capable of inhibiting allergic inflammation caused by Th2 and Th17 and restoring the immune system [38]. Inflammatory cytokines are produced by Th2 subtype effector T lymphocytes (Th2s) once they have been activated, such as IL-4, IL-5, and IL-13, and finally leads to the enhancement of IgE, which induces the aggravation of AD. Compared to healthy people, AD patients have significantly lower levels of short-chain fatty acids (SCFAs) such as propionate and butyrate [39, 40]. Propionate and butyrate are important short-chain fatty acids that may inhibit inflammation in AD patients who are prone to Th2-induced and Th17-induced inflammation. An analysis of fecal samples from AD patients performed in South Korea found that the abundance of Faecalibacteriumprausnitzii was decreased compared with that of the healthy control group, while the production of SCFA was also significantly reduced [39–41]. Propionate in the intestinal tract is mainly produced by Dialister [42], fibers can be decomposed by Prevotella, and propionate and butyrate are produced [39]. Specifically, both of them inhibit histone deacetylase [43] and induce peripheral CD4$^+$T cells to differentiate into Treg cells, IL-10 is then produced, inhibiting Th2 and Th17 cell function [44]. According to our MR analysis, Dialister and atopic dermatitis are inversely related. Therefore, compared with the speculation in the previous literature, we have confirmed through this study that the protective effects of these bacterial groups on AD are related to SCFAs, especially Dialister. Furthermore, we identified some other bacterial groups that may be associated with AD, which has provided a definite guiding value for the future probiotic treatment of AD.

It has been reported that dysbiosis of gut microbiota and abnormal metabolic pathways affect the pathophysiology of acne. These include: Sterol regulatory element binding protein 1 (SREBP-1), sebum fatty acids, and sebum triglycerides are stimulated by nutrient signal interruption, resulting in the proliferation of Propionibacterium acnes [45]. In addition, some patients with acne have also been found to have low gastric acid levels, that is, they suffer from perchloric acidemia. Bacteria from the colon migrate to the small intestine when acidity levels are low, resulting in an imbalance of the gut microbiota and the growth of intestinal bacteria, thereby increasing intestinal permeability and promoting skin inflammation [46]. A survey has shown a significant reduction in the number of Lactobacilli in people with acne vulgaris [47]. Research suggests that Bacteroides may promote the development of acne by degrading polysaccharides, potentially enhancing inflammatory responses, stimulating angiogenesis, and weakening immune defenses. This is consistent with our MR study results. Additionally, we identified other genera that may be associated with the gut microbiota, such as Allisonella, Coprococcus3, among others. This also provides direction for further research.

At present, there are few research reports on gut microbiota and lichen planus. Lichen planus is a chronic inflammatory skin disease. By consulting the existing literature, we have only found the correlation between oral lichen planus and gut microbiota, and no other types of reports in the traditional sense. A study has shown that the dysbacteriosis of the oral bacterial flora contributes to the occurrence of the disease [11, 48]. This is the first MR analysis to assess whether genetically predicted gut microbiota risks will subsequently affect lichen planus, and our results have revealed some flora and lichen planus may be causally related, which may be related to metabolites of the gut microbiota as well as immune effects. The exact mechanism, however, remains unclear, and further research is needed. It also demonstrated the existence of the gut-skin axis, highlighting the interconnectedness and feedback between the two, maintaining overall health. Disruption of this balance may contribute to the development of certain diseases.

Several advantages are associated with this study. Firstly, it is the first bidirectional TSMR study to link gut microbiota to psoriasis, atopic dermatitis, acne, and lichen planus, and

confounding factors or reverse causal relationships do not affect it. Secondly, we set strict conditions for screening instrumental variables, and the causal relationships obtained through rigorous analysis and exclusion of the results can only be considered valid. Thirdly, we provide genetic evidence for an intestinal-cutaneous axis study. TSMR analysis was used to identify correlations with PSO, AD, acne, and LP. Related 29 bacterial groups. These identified important bacterial groups can be used as candidate microbiota interventions in future clinical trials for immunological diseases. At the same time, our results to study immune skin diseases may provide a new perspective: targeted regulation of specific bacterial groups, by inhibiting the growth of harmful bacteria and supplementing beneficial bacteria, prevention and treatment of associated skin conditions could be achieved.

The study also has some limitations. First, gut microbiota GWAS statistics are limited in instrumental variables, and species-level data aren't available. Second, the original study lacks demographic data; There are still many clinical and basic studies that need to be conducted in order to validate our results. We were unable to conduct subgroup analyses based on factors such as gender.

As for the relatively small number of corrected FDR results, we considered that it might be since the P values of the main MR analysis results were mostly concentrated in the range of 0.9–1 and were not uniformly distributed, which would result in an excessive correction, so the original results were also of reference significance.

## Supporting information

**S1 Table. All SNPs information used as IVs.**
(CSV)

**S2 Table. Gut microbiota and MR results for PSO.**
(XLSX)

**S3 Table. Gut microbiota and MR results for AD.**
(XLSX)

**S4 Table. Gut microbiota and MR results for ACNE.**
(XLSX)

**S5 Table. Gut microbiota and MR results for LP.**
(XLSX)

## Acknowledgments

Thanks to FinnGen participants and researchers, as well as the MiBioGen consortium for providing GWAS data on the gut microbiota.

## Author Contributions

**Data curation:** Fei Feng.

**Investigation:** Fei Feng.

**Methodology:** Fei Feng.

**Supervision:** Xueyi Wu.

**Visualization:** Fei Feng.

**Writing – original draft:** Fei Feng, Ruicheng Li, Rui Tian.

**Writing – review & editing:** Xueyi Wu, Nannan Zhang, Zhenhua Nie.

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
