## [Decision Letter · Decision Letter 0]

8 Dec 2023

PONE-D-23-25279The causal relationship between gut microbiota and immune skin diseases: a bidirectional Mendelian randomizationPLOS ONE

Dear Dr. feng,

Thank you for submitting your manuscript to PLOS ONE. After careful consideration, we feel that it has merit but does not fully meet PLOS ONE’s publication criteria as it currently stands. Therefore, we invite you to submit a revised version of the manuscript that addresses the points raised during the review process.

We look forward to receiving your revised manuscript.

Kind regards,

Donatella Mentino

Academic Editor

PLOS ONE

Journal Requirements:

2. Please amend the manuscript submission data (via Edit Submission) to include author Xueyi Wu and Nannan Zhang.

Additional Editor Comments :

In your work why have you only dealt with some skin pathologies?

explain the discussion and conclusion better

Reviewers' comments:

Reviewer's Responses to Questions

**Comments to the Author**

1. Is the manuscript technically sound, and do the data support the conclusions?

Reviewer #1: Yes

2. Has the statistical analysis been performed appropriately and rigorously? 

Reviewer #1: Yes

3. Have the authors made all data underlying the findings in their manuscript fully available?

Reviewer #1: Yes

4. Is the manuscript presented in an intelligible fashion and written in standard English?

Reviewer #1: Yes

5. Review Comments to the Author

Reviewer #1: This is a bidirectional two-sample mendelian randomisation study that appraised the causal relationship between gut microbiota and skin diseases.The manuscript was well written, and analyses were comprehensively conducted. Nevertheless, the quality of this manuscript may be improved by addressing the following issues.

Major:

1. The authors extracted GWAS summary statistics for skin diseases from the FinnGen study (data freeze 5) which was released in 2021 in 218,792 individuals. However, the most up-to-date datasets were released in April 2023, with much larger sample sizes (377,277 individuals). I would like to see the results using the most updated datasets.

2. I suggest calculating the statistical power for each MR analysis.

3. The formula to calculate F statistics seems wrong and the citation of references is confusing in section 2.1 line 111-114, please carefully check and revise these.

Minor:

1. The colomns "mr_keep.exposure","pval_origin.exposure" and "id.exposure" are not necessary in supplementary table 1, please remove them. Since the "eaf.exposure" is not available for each SNP, please also remove it.

2. The colomns "id.exposure" and "id.outcome" are not necessary in supplementary tables 2-4, please remove it.

3. Please add P values for heterogeneity and pleiotropy from the IVW and MR Egger methods to supplementary tables 2-4.

4. The authors used different terms including "gut microbiota", "gut flora", and "intestinal flora" in differnet places, please use "gut microbiota" throughout the whole manuscript.

6. PLOS authors have the option to publish the peer review history of their article (what does this mean?). If published, this will include your full peer review and any attached files.

Reviewer #1: No

---

## [Author Response · Author response to Decision Letter 0]

28 Dec 2023

Dear Professor Donatella Mentino,

We express our gratitude to you and the expert reviewers for the thorough examination and approval of our manuscript titled "The causal relationship between gut microbiota and immune skin diseases: a bidirectional Mendelian randomization" by Fei Feng et al. In this study, we utilized Mendelian randomization to explore the relationship between gut microbiota and several skin diseases. The results revealed a causal relationship between various microbial groups in the gut and conditions such as psoriasis, atopic dermatitis, acne, and lichen planus. Changes in the abundance of these microbial groups may contribute to the onset and progression of these skin disorders. Our findings underscore the significance of the gut-skin axis in clinical contexts and provide new insights into the treatment of skin diseases.

We have carefully addressed the reviewers' suggestions, offering a point-to-point response to their comments. Enclosed, please find the revised manuscript with changes highlighted, along with a "clean" copy formatted according to the journal's requirements. We believe that these revisions, guided by the reviewers' critiques and Manuscript Edits, have substantially enhanced the quality of the manuscript, and we hope it meets the standards for publication in PLOS ONE.

We appreciate your consideration in allowing us to revise our manuscript and eagerly anticipate your feedback.

Sincerely,

Fei Feng

We thank the editor and the reviewers for their thorough examination of our manuscript, and the valuable and insightful comments provided. We have revised the manuscript accordingly and provided a point-to-point response to reviewers’ comments below.

Editor Comments:

In your work why have you only dealt with some skin pathologies?

Author Response:

In the manuscript, we focused on a subset of skin diseases, acknowledging a limitation in our coverage. Due to constraints related to time, resources, and the research direction of our team, we primarily addressed several common skin conditions encountered in our clinical work. This approach facilitates a more in-depth understanding and treatment approach to these diseases. Additionally, we recognize the potential for further clinical studies to validate the obtained results.

Reviewers' comments:

Reviewer #1：

Major:

1. The authors extracted GWAS summary statistics for skin diseases from the FinnGen study (data freeze 5) which was released in 2021 in 218,792 individuals. However, the most up-to-date datasets were released in April 2023, with much larger sample sizes (377,277 individuals). I would like to see the results using the most updated datasets.

Author Response:

Thank you very much for your suggestions. We have realized that the data we initially used were from the FinnGen study (data freeze 5), which significantly differed in scale from the latest release. Consequently, we conducted a new Mendelian randomization (MR) analysis using the most recent data available as of December 18, 2023 (data freeze 10, 412,181 individuals). The results have been appropriately adjusted in the manuscript. With the increase in sample size, some findings from the initial analysis are no longer statistically significant in the second analysis. We believe that the results from the larger sample size are more reliable. We appreciate your guidance, which has contributed to the rigor and completeness of our study.

2. I suggest calculating the statistical power for each MR analysis.

Author Response:

Thank you very much for your advice. Indeed, we have considered the power for each Mendelian randomization (MR) analysis. However, the GWAS raw data for the gut microbiota we used lack the values for effect allele frequency (EAF) and minor allele frequency (MAF). Consequently, we were unable to calculate the power. We then supplemented the EAF values based on human "1000g" genomic data and computed the power accordingly. These results will be included in our supplementary files. While this approach enhances rigor, the authenticity and accuracy of the supplemented data may be subject to discussion. Therefore, we opted not to present these details in the main manuscript.

3. The formula to calculate F statistics seems wrong and the citation of references is confusing in section 2.1 line 111-114, please carefully check and revise these.

4. Author Response:

I deeply apologize for errors made during the writing process. The citation issues for references 111-114 have been corrected. Regarding the calculation formula for F statistics, we explored various methods from relevant literature. The classic formula is F = R2*(N-K-1)/K*(1-R2), where R2 = 2β2 *(1-MAF) *MAF. However, due to the lack of EAF and MAF values in the original data, implementing this method proved challenging. Consequently, we opted for an alternative formula based on other literature, specifically F = β2 exposure/SE2 exposure.

Minor:

1. The columns "mr_keep.exposure","pval_origin.exposure" and "id.exposure" are not necessary in supplementary table 1, please remove them. Since the "eaf.exposure" is not available for each SNP, please also remove it.

2. The columns "id.exposure" and "id.outcome" are not necessary in supplementary tables 2-4, please remove it.

3. Please add P values for heterogeneity and pleiotropy from the IVW and MR Egger methods to supplementary tables 2-4.

4. The authors used different terms including "gut microbiota", "gut flora", and "intestinal flora" in different places, please use "gut microbiota" throughout the whole manuscript.

Author Response:

Thank you very much for your guidance on the shortcomings of the manuscript. We have addressed each of the points you highlighted in the new draft, and these suggestions have contributed to making the paper more rigorous and scientifically sound.

---

## [Editor Report · Decision Letter 1]

25 Jan 2024

The causal relationship between gut microbiota and immune skin diseases: a bidirectional Mendelian randomization

PONE-D-23-25279R1

Dear Dr.fei feng

We’re pleased to inform you that your manuscript has been judged scientifically suitable for publication and will be formally accepted for publication once it meets all outstanding technical requirements.

Kind regards,

Donatella Mentino

Academic Editor

PLOS ONE
---

## [Editor Report · Acceptance letter]

8 Mar 2024

PONE-D-23-25279R1 

PLOS ONE

Dear Dr. Feng, 

I'm pleased to inform you that your manuscript has been deemed suitable for publication in PLOS ONE. Congratulations! Your manuscript is now being handed over to our production team.

Kind regards, 

on behalf of

Dr. Donatella Mentino 

Academic Editor

PLOS ONE